# The Relationship between Wellbeing, Self-Determination, and Resettlement Stress for Asylum-Seeking Mothers Attending an Ecosocial Community-Based Intervention: A Mixed-Methods Study

**DOI:** 10.3390/ijerph20227076

**Published:** 2023-11-17

**Authors:** Yufei Mandy Wu, Jens Kreitewolf, Rachel Kronick

**Affiliations:** 1Division of Social and Transcultural Psychiatry, Department of Psychiatry, Faculty of Medicine and Health Sciences, McGill University, Montreal, QC H3A 1A2, Canada; rachel.kronick@mcgill.ca; 2Department of Psychology, Toronto Metropolitan University, Toronto, ON M5B 2K3, Canada; 3Department of Psychology, McGill University, Montreal, QC H3A 1G1, Canada; 4Department of Mathematics and Statistics, McGill University, Montreal, QC H3A 1G1, Canada; 5Lady Davis Research Institute, Jewish General Hospital, Montreal, QC H3T 1E2, Canada; 6Institut Universitaire SHERPA, Montreal, QC H3N 1Y9, Canada

**Keywords:** self-determination theory, wellbeing, resettlement stress, self-efficacy, asylum seeker, mothers

## Abstract

Psychosocial support programs have been increasingly implemented to protect asylum seekers’ wellbeing, though how and why these interventions work is not yet fully understood. This study first uses questionnaires to examine how self-efficacy, satisfaction of basic psychological needs, and adaptive stress may influence wellbeing for a group of asylum-seeking mothers attending a community-based psychosocial program called Welcome Haven. Second, we explore mothers’ experiences attending the Welcome Haven program through qualitative interviews. Analysis reveals the importance of relatedness as a predictor of wellbeing as well as the mediating role of adaptive stress between need satisfaction and wellbeing. Further, attending Welcome Haven is associated with reduced adaptive stress and increased wellbeing, which correspond with the thematic analysis showing that attendance at the workshops fostered a sense of belonging through connection with other asylum seekers and service providers as well as empowerment through access to information and self-expression. The results point to the importance of community-based support that addresses adaptive stress and the promotion of social connection as key determinants of wellbeing. Nonetheless, the centrality of pervasive structural stressors asylum seekers experience during resettlement also cautions that relief offered by interventions may be insufficient in the face of ongoing systemic inequality and marginalization.

## 1. Introduction

Asylum seekers, also known as refugee claimants, are people who arrive in a host country seeking protection from harm or persecution in their country of origin and must undergo an often-protracted process to determine if they meet criteria for refugeehood. When compared to refugees, that is, those who have been granted protection in a host country, asylum seekers have a higher prevalence of psychiatric morbidity thought to be linked with the specific and especially high burden of post-migratory stressors for claimants, including uncertainty itself [1,2]. Among asylum seekers, mothers have unique challenges specific to migrating as part of a family and their gendered and racialized identities as caregivers. Like for all claimants, mothers’ specific stressors often intersect with other resettlement challenges such as social isolation, poverty, and racism, producing and perpetuating health inequities [3,4].

### 1.1. Aims

This study is nested in a larger investigation examining the implementation of a community-based psychosocial support program for asylum-seeking families in Montreal, QC, called Welcome Haven. The Welcome Haven takes place once a week in rotating community centres as a drop-in support program, connecting families to resources and information and providing a space for creative expression for children. While fathers often attend, children are predominantly accompanied by mothers at our drop-in workshops. Previous research on social support programs for refugees or asylum seekers has, for the most part, looked at concrete psychosocial or social outcomes separately, for example, financial self-sufficiency, empowerment, or social connections [5]. Few, if any, studies have integrated psychological models examining the relationship *between* psychological factors that might predict the flourishing of asylum-seeking mothers in resettlement. We wanted to better understand the experiences of these mothers and to develop empirical support for the Welcome Haven intervention that could point to underlying mechanisms of efficacy. Imperative to this endeavor is modelling how multiple psychosocial constructs interact to shape asylum seekers’ subjective wellbeing. Thus, we set out in this study to investigate how mothers’ sense of *agency*, *belonging*, and *resettlement stress* are related to their *wellbeing*. We also aim to understand how mothers understand these constructs (of agency, belonging, wellbeing, and resettlement stress) and their perceptions of the impact of the program on their wellbeing. To study this, we draw on three psychological conceptualizations, not previously studied together for asylum seekers, that, respectively, capture the variables of agency, belonging, and resettlement stress. Below, we elucidate these three constructs (self-efficacy, basic psychological need satisfaction, and adaptive stress) and outline the components of community-based psychosocial support.

### 1.2. Self-Efficacy and Basic Psychological Needs Theory

Two commonly studied psychological concepts associated with wellbeing are self-efficacy from social cognitive theory [6] and basic psychological needs theory from self-determination theory (SDT) [7,8]. We start with an outline of both theories and how they apply to the displaced population, followed by a comparison that highlights the differences between the two concepts.

#### 1.2.1. Self-Efficacy

When applied to trauma-affected individuals, self-efficacy is described as “the perceived capability to manage one’s internal functioning and the myriad environmental demands of the aftermath occasioned by a traumatic event.” [9] (p. 1130) Closely connected with the notion of agency, a generalized sense of self-efficacy is a belief in one’s overall competence to deal with a variety of stressful situations [10,11]. Self-efficacy is a predictor of mental health [12,13]. For refugees, high self-efficacy has been found to foster resilience [14] and has been associated with lower psychological distress and a greater subjective sense of wellbeing [12]; fewer depressive and anxiety symptoms [15]; and higher psychological security [16]. Tip et al. [13] also found that higher self-efficacy could lead to more positive affect longitudinally amongst resettled refugees.

#### 1.2.2. Basic Psychological Needs Theory

Basic psychological need theory, as part of SDT, posits that (i) autonomy (one’s need to self-regulate their experiences and actions); (ii) competence (one’s need to feel effective and masterful); and (iii) relatedness (one’s need to feel social connectedness) are the three basic psychological needs to facilitate human flourishing [17]. For asylum seekers, these needs are often thwarted. For instance, the uncertainty and powerlessness they experience as they await their refugee determination may threaten their autonomy. The lack of acknowledgment of their professional degrees, deprivation of resources and barriers to service access may thwart their sense of competence. The loss of kinship networks and discrimination in the host society may diminish their sense of social connectedness [18,19]. Considering that these psychological needs are fundamental, limited need satisfaction may lessen wellbeing and increase vulnerability to psychopathology [7,8]. Indeed, there is evidence that the satisfaction of basic psychological needs is positively correlated with resilience and post-traumatic growth among traumatized individuals [20,21] and that refugees who experience more need frustration exhibit more symptoms of depression and post-traumatic stress disorder [22]. Psychosocial interventions designed to address these needs showed promise in reducing symptoms of depression and generalized stress among refugees [22]; thus, researchers have advocated for policies and interventions to focus on fulfilling forcibly displaced people’s psychosocial needs for autonomy, competence, and relatedness during resettlement [19].

While both self-efficacy and need satisfaction are known predictors of asylum seekers’ psychosocial wellbeing, self-efficacy has been more extensively researched in predicting wellbeing among refugees and asylum seekers compared to basic psychological needs theory. However, there are important differences between the two constructs. First, self-efficacy is mainly conceptually related to autonomy and competence needs [7]. Relatedness is not directly captured in the measurement of self-efficacy. Second, the basic psychological needs theory accounts for one’s autonomous values and desires that drive action and wellbeing compared to self-efficacy, which proposes that the capacity to act is more important than autonomy over the decisions to act [23]. In this sense, the values and desires that underpin an individual’s sense of autonomy are only captured in measurements of need satisfaction but not self-efficacy. For asylum seekers, this could mean that they participate in activities that do not align with their values; even though they might experience a high level of self-efficacy, those activities would not necessarily enhance their wellbeing.

Based on the differences between these two constructs of self-efficacy and need satisfaction, we hypothesize that need satisfaction will be a more significant predictor of asylum seekers’ wellbeing than self-efficacy.

### 1.3. Adaptive Stress

Unlike self-efficacy and need satisfaction, which are applicable to the general population, the Adaptive Stress Index (ASI) captures resettlement stress that specifically affects asylum seekers. The ASI was based on the ADAPT model, an ecosocial framework that postulates that there are five psychosocial pillars affected by mass conflict and displacement, including safety/security, bonds/network, justice, roles and identities, and existential meaning [24]. The restoration of these pillars is understood as critical to asylum seekers’ capacity to recover [25]. Based on this conceptual foundation, Tay et al. [26] developed a scale called the Adaptive Stress Index (ASI), which encapsulates a series of adaptive stressors that “arise from disruptions to key psychosocial support structures” that asylum seekers experience (p.1). Considering that post-migratory stressors were found to be much more significant in predicting mental health problems compared to self-efficacy [27], adaptive stress warrants examination as a potential determinant of wellbeing in addition to self-efficacy and need satisfaction. Adaptive stress may also be associated with the basic psychological needs proposed in SDT. For example, because of language barriers, cultural adaptation, and a disrupted sense of continuity of life, mothers likely experience changes in their level of autonomy and competence. Similarly, the disruption or deprivation of one’s role in familial and communal relationships would affect their sense of relatedness. At the same time, if claimants’ needs are satisfied, they should also experience less adaptive stress in their role and existential meaning. We thus hypothesize that asylum-seeking mothers’ satisfied basic psychological needs will predict reduced adaptive stress, which will in turn positively impact asylum-seeking mothers’ subjective wellbeing.

### 1.4. Community-Based Psychosocial Support

Community-based psychosocial support programs are increasingly implemented to protect asylum seekers’ wellbeing [5,25,28,29,30,31]. These programs are distinct from other mental health services because of their emphasis on facilitating social connectedness and service outreach, delivered through various interventions delivered in a wide array of organizational structures [32,33]. We define community-based psychosocial supports using the Interagency Standing Committee (IASC) guidelines for intervention in emergency settings. IASC outlines a pyramid of intervention levels, starting with redressing basic safety and security needs of the population and ending with specialized mental health services [34]. In between these levels of intervention is community-based psychosocial intervention, which aims at strengthening community and family supports and also non-specialized person-to-person supports (See Figure 1). Unlike individualistic and expensive specialized mental healthcare services, community-based psychosocial support programs take an ecological approach [35] that attends to both individual and collective stressors. A better understanding of these psychosocial interventions may help identify the critical components that impact wellbeing. This is especially relevant for asylum-seeking mothers with children, considering the unique sociocultural barriers they face in accessing specialized mental healthcare [36,37,38,39] and the importance of social support and inclusion in addressing their psychosocial needs [40,41,42,43].

### 1.5. Research Objectives

This mixed-methods study will (i) measure the predictive power of self-efficacy and satisfaction of basic psychological needs (including the need for autonomy, competence, and relatedness) as well as adaptive stress and asylum-seeking mothers’ subjective wellbeing during resettlement as participants in a community-based psychosocial support program and (ii) describe mothers’ perceptions of how attending the program shaped their self-efficacy, self-determination, adaptive stress, and wellbeing.

For the quantitative component, we hypothesize that, first, the satisfaction of basic psychological needs would predict adaptive stress and wellbeing better than self-efficacy alone. Second, in exploring the relationship between need satisfaction and adaptive stress, we hypothesize that adaptive stress would serve as a mediator of the effect of need satisfaction on wellbeing, considering that not only would need satisfaction be a significant predictor of both adaptive stress and wellbeing but also that adaptive stress would be a significant predictor of wellbeing. Our qualitative inquiry examines how attending a community-based psychosocial program may shape mothers’ experiences of their self-efficacy, self-determination, adaptive stress, and wellbeing. Better understanding the relationship between these psychosocial factors and how needs may be met through psychosocial interventions, is essential for development of interventions that will enhance the wellbeing of claimants during resettlement.

## 2. Methods

### 2.1. Study Design and Sample

The study was conducted as part of a two-year community-based, multi-site, psychosocial program for asylum-seeking families in Montreal, Québec, called “Welcome Haven” or ‘Espace Le Havre’. Following ethics approval from CIUSSS in September 2021, the program was implemented in October 2021 in collaboration with local community organizations and the regional program supporting the settlement and integration of asylum seekers. The main program consists of in-person, three-hour drop-in sessions for refugee claimant families held on a weekly basis. The sessions were designed to offer both practical and emotional support to families through information or arts-based workshops, both for parents and children. Information-based workshops focused on inviting service providers with different expertise in resettlement to impart knowledge on asylum seekers’ rights and service access. Arts-based workshops, on the other hand, were led by trained therapists and centred on creative self-expression. Additionally, a shared communal meal began each workshop to foster community belonging.

This mixed-methods study followed a convergent design, with quantitative and qualitative data independently collected and analyzed. The results were subsequently merged to “advance multiple perspectives […] validate one database with the other” [44] (p. 36). In analyzing quantitative and qualitative data, we were seeking a more comprehensive portrait of the active ingredients of this community-based psychosocial program than either data set could provide alone. Especially for asylum seekers, who are a “hard-to-reach” and generally understudied population, quantitative data collected in this study could offer preliminary insights into the relationship between self-efficacy, need satisfaction, resettlement stress, and subjective wellbeing from an etic (or outside observer) perspective. Qualitative data provide an emic (or insider) perspective by giving us first-person perspectives on ‘what works’ for mothers. In our analysis, the quantitative and qualitative data were used to make sense of each other, thus enhancing validity and authenticity [44]. Specifically, we use our quantitative analysis to shed light on *what* is happening, and the qualitative results illuminate *how* this is happening. Through a comparison of the two, we may point out discrepancies and coherence between subjective accounts and empirical ones. 

Quantitative data were collected through individually administered questionnaires that gathered sociodemographic information and measures of subjective wellbeing, need satisfaction, self-efficacy, and adaptive stress. Qualitative data were collected through semi-structured interviews conducted with participants who also completed the questionnaire, which solicited participants’ narratives and perspectives on resettlement, agency, and experiences attending the workshop.

Following a cross-sectional design, the participants of the quantitative component were conveniently sampled from the participants of the Welcome Haven program. The inclusion criteria were (1) being at least 18 years old, (2) awaiting refugee status in Canada, (3) having participated in at least one workshop, and (4) identifying as a mother. The first author, along with an interpreter when needed, approached all participants who met the inclusion criteria during the workshop for an informed consent discussion. Participation was voluntary. In most cases, questionnaires were completed in paper-and-pencil format with the first author, and interpreters were available when an explanation was needed. The first author also checked the questionnaire for missing responses after completion and followed up with participants to complete those items if they were unintentionally missed. Due to the public health regulation in Quebec from December 2021 to March 2022, five participants recruited during that time chose to complete the questionnaire via a secured link online. Certain scales included in the questionnaires were originally in English, French, and Spanish. For the scales that were not available in French and Spanish, two trained research assistants with proficiency in the language and experience working with asylum seekers used a back-and-forth method to translate the original English questionnaire, with inconsistencies discussed and reconciled.

For the qualitative component of the study, participants for the interviews were purposively sampled from those who had completed the questionnaire, with an additional inclusion criterion of having participated in at least two Welcome Haven workshops. When participants spoke only Spanish, professional or trained volunteer interpreters were used during the interviews. The first author conducted all interviews with English-speaking participants and Spanish-speaking participants with the assistance of an interpreter, and two other research assistants conducted interviews with French-speaking participants. The majority of interviews were audio-recorded following participant consent, and in instances where the participant declined to be recorded, the first author endeavored to transcribe as much of the interview verbatim as possible. All English and French interviews were subsequently transcribed in the language of the interview. Spanish interviews were transcribed either in English based on live interpretation or in both Spanish and English interpretation to ensure accuracy.

### 2.2. Measurements

Questionnaires gathered sociodemographic information (including age, region of origin, education level, length of resettlement in Canada, marital status, number of children, and number of workshops attended) and measures of subjective wellbeing, satisfaction of basic psychological needs, self-efficacy, and adaptive stress. All scales employed in this study have been cross-culturally validated and have been used with a refugee or asylum-seeking population [12,13,25].

Subjective wellbeing (SWB) was assessed using the cognitive subscales of the Satisfaction With Life Scale [45] and the cognitive subscale of the Mood Report [46], which assesses emotional wellbeing. Cognitive wellbeing refers to an overall assessment of life, whereas emotional wellbeing refers to the frequency of experiencing positive and negative emotions [47]. A composite index of SWB was calculated with the mean scores of positive affect, reversed negative affect, and satisfaction with life. Need satisfaction was assessed with the Basic Psychological Need Satisfaction Scale [48], shortened and adapted by Holding et al. [49]. Self-efficacy was measured with a shortened version of the General Self-Efficacy Scale [11], adapted by Tip et al. [13] to use with resettled refugees in the UK. Adaptive stress during resettlement was assessed using two of the subscales (role/identity and existential meaning) from the Adaptive Stress Index (ASI) [26], which were most relevant to our study. More information on each scale, including specific items used, the Likert scale, and Cronbach’s alpha, can be found in Table 1.

### 2.3. Data Analysis

SPSS version 29.0 (IBM Corp, New York, NY, USA) and Jamovi version 2.3.21.0 (Jamovi, Sydney, Australia) were used to conduct all statistical analyses. All 40 participants were included in the data analysis with no imputation since the missing data were less than 5%. All control variables and predictors were standardized (z-scored) before the correlation and hierarchical linear regression analyses were conducted, a necessary procedure for calculating composite scores with different scales. Hierarchical linear regression analysis was computed with the enter method. Two three-stage hierarchical multiple regressions were conducted with SWB and resettlement stress as the dependent variables. Sociodemographic variables including age, highest level of completed education, civil status, number of children, months of resettlement, and program attendance (number of visits to Welcome Haven) were entered at stage one of the regression as control variables. Self-efficacy was entered at stage two, and need satisfaction was entered at stage three. Based on the preliminary analyses, we excluded the variables (age, education, civil status, and number of children) unrelated to the outcomes from the central analyses. The results of the preliminary analyses can be found in the supplementary materials in a file titled “Appendix A: Preliminary Analyses”.

All assumptions for conducting regression analyses, including the normal distribution of residuals, autocorrelation of residuals (Durbin–Watson test), and homoscedasticity, were met for both dependent variables, subjective wellbeing, and adaptive stress (although the Durbin–Watson score was slightly lower at 1.2 for subjective wellbeing). None of the predictors showed multicollinearity (here defined as VIF > 2.7). The mediation effect of resettlement stress on the relationship between need satisfaction and wellbeing was tested with a simple mediation analysis using the “medmod” package in Jamovi. Besides a large sample z-test of the mediation effect, a bootstrapping estimation approach with 5000 samples was also used to calculate the standard errors. The statistical significance cutoff point was set at α = 0.05.

For qualitative data analysis, transcripts were entered into a qualitative data management software (NVivo). Because the first author was English-speaking, all interviews were translated with reliable online software into English for data analysis. In the cases where the translation was suspected to be inadequate, the first author verified the meaning of the translation with the corresponding interviewer to improve accuracy. Coding was then conducted inductively following principles of thematic content analysis as described by Braun and Clarke [50]. The codes were then iteratively reviewed and refined, and themes were generated and defined with interpretations that aim to address conceptualizations of self-efficacy, self-determination, adaptive stress, and wellbeing.

## 3. Results

### 3.1. Quantitative Results

In total, 42 participants consented to participate in the study, 40 of which completed the questionnaire; 2 participants’ responses were discarded due to incompletion. Table 2 includes the sociodemographic characteristics of this sample. Two additional participants’ responses were excluded from regression analyses because of missing values.

In the first regression analysis, both *model* 2 and *model* 3 (the full model) explained a significant proportion of variance in subjective wellbeing. For *model* 2, adjusted *R*^2^ = 0.182, *F* (3, 34) = 3.751, *p* = 0.020; for *model* 3, adjusted *R*^2^ = 0.310, *F* (4, 33) = 5.156, *p* = 0.002. The F statistic increased significantly from *model* 1 to *model* 2 (*F_change_* (1,34) = 3.643, *p* = 0.031), with an adjusted *R*^2^ change of 0.095. There was also a significant increase in the F statistic from *model* 2 to *model* 3 (*F_change_* (1,33) = 7.289, *p* = 0.011), with an adjusted *R*^2^ change of 0.128. The models thus showed that self-efficacy no longer significantly predicted subjective wellbeing when need satisfaction was accounted for. Additionally, we found that months of resettlement (ß = −0.473) and program attendance (ß = 0.512) significantly predicted wellbeing in *model* 3. Table 3 shows the result of this analysis.

In the second regression analysis, *model* 3 (the full model) explained 29.8% of the variance in adaptive stress, (*F* (4, 32) = 4.827, *p* = 0.004). The F statistic increased significantly from *model* 2 to *model* 3 (*F_change _
*(1,32) = 13.245, *p* < 0.001), with an adjusted *R*^2^ change of 0.26, indicative of the significant effect of need satisfaction, but not self-efficacy, in predicting adaptive stress. Additionally, program attendance was found to be a significant predictor of adaptive stress in *model* 3, ß = −0.557. Table 4 shows the result of this analysis.

Figure 2 depicts the simple mediation model examining whether the effect of satisfaction on wellbeing is mediated by adaptive stress. The results indicate that need satisfaction negatively predicted adaptive stress (‘a’ path, *B* = −0.233, *SE* = 0.065, *Z* = −3.57, *p* < 0.001) and that adaptive stress negatively predicted wellbeing (‘b’ path, *B* = −0.868, *SE =* 0.286, *Z* = −3.04, *p* = 0.002). Consistent with a full mediation effect, need satisfaction was no longer a significant predictor of wellbeing after controlling for the mediator of adaptive stress. The direct path was not significant (‘c’ path, *B* = 0.173, *SE* = 0.136, *Z* = 1.27, *p* = 0.205), but the indirect path was (‘c’ path, *B* = 0.20, *SE* = 0.08, *Z* = 2.43, *p* = 0.015). These findings suggest that need satisfaction increases subjective wellbeing through reduced adaptive stress.

### 3.2. Qualitative Results

The demographic information of the 13 interview participants can be found in Table 5. In our conversations with asylum-seeking mothers about their experiences of self-efficacy, agency, and resettlement stress and how attending the Welcome Haven program may have shaped these experiences, three central themes emerged: (1) community and relatedness, (2) competence and empowerment, and (3) enhanced wellbeing.

### 3.3. Community and Relatedness

Asylum-seeking mothers frequently referred to the sense of community and social connections that they experienced at Welcome Haven as the main reason for their participation. A sense of community at Welcome Haven was facilitated through bonding with other asylum seekers and connection with facilitators that participants reported fostered a sense of belonging.

#### 3.3.1. Bonding with Other Asylum Seekers

Several participants emphasized the importance of being able to connect with other asylum seekers, not only for informational support such as exchanging knowledge on the asylum claim process or sharing parenting techniques that facilitated children’s adjustment but also for emotional support from a shared understanding of the difficulties of resettlement. Adriana gave the example of connecting with another mother from a different cultural and racial background:


*When I speak up, she says I feel what you feel. [….] It’s necessary for people like us who don’t know anybody […] to meet people who are in the same situation as you.*


Importantly, for most mothers, connection with other participants was possible even when there were language barriers. Daniela explained that “When I was first at the workshop, I felt free, that I’m with people again, even though they don’t understand me, I felt that I’m with people.” This mother had experienced severe isolation in a temporary shelter when her family first arrived in Canada. For Fernanda, who described herself as “very social”, the workshops gave her an opportunity to develop that more “open and sociable” side of herself, a personality she took pride in. While some participants acknowledged their preference to connect with people from their own cultural background, many still appreciated learning “stories of [different] countries and cultures” (Rosine).

#### 3.3.2. Connection with Service Providers

Many mothers expressed gratitude for their connection with facilitators, who were perceived as host society members but also had lived experience of migration. Facilitators were seen as providing instrumental support. For example, Mary recounted her experience with a child psychiatrist (the principal investigator) who approached her at the workshop to discuss the service options for her son with special needs and who later facilitated easier access to these services that she had tried but failed to access for months. Divine shared how touched she was when a social worker at the workshop told her “you can call me anytime” and subsequently assisted her with apartment hunting. Cassandra explained that service providers from other organizations were more “impersonal” and that the directions they gave still left her feeling “lost” because her family’s needs extended beyond basic instrumental support such as receiving welfare and finding winter clothes. Instead, the information and support provided at Welcome Haven aimed to address the complex post-migration needs and was extolled by one participant as having helped her “through many stages in [her] life” when she was all alone.

Several mothers emphasized the emotional support they received from the service providers at the workshop that fostered a sense of belonging, calling Welcome Haven their “second family”. When Aisha needed to express herself, a service provider “just listened” and gave her a hug despite social distancing rules. Mary shared that she did not have any family members to talk to, but because she could share her worries and concerns with the service providers at Welcome Haven, her “loneliness would go away.” Even small gestures like a phone call invitation were significant, as Tenneh explained: “The first [time] you called me, you [told] me: ‘we need you, you are important.’ For the first time someone called me like I’m important, I feel good.” Similarly, Gloria described appreciation for being “well-received” the first time she attended the workshop:


*People I didn’t know approached me, it’s wonderful to feel that wow, I have people around me. I’m not alone. People who are going to make it easier for me to fit in, I’m not just another commodity, just another object. One more workforce. I am a person who is considered.*


For this claimant, connection with service providers at Welcome Haven contrasted her experience of being dismissed as another “object” or “workforce”. Considering the status-based discrimination and exclusion many claimants endured, a welcoming experience at the workshops signaled acceptance by host society members. For participants, the growing sense of belonging appeared to bolster their sense of self-efficacy. For example, for Fatima, who was a specialist physician in her country of origin, attending and subsequently volunteering at Welcome Haven gave her the opportunity to integrate professionally and “be part of the research world,” which gave her a sense of competence and purpose.

### 3.4. Competence and Empowerment

Both the information-based and arts-based workshops at Welcome Haven were experienced as empowering by the participants. Several participants brought up examples of how the knowledge they learnt from the workshops positively impacted their resettlement. One example mentioned by Aisha was the tenants’ rights workshop, in which she was able to attain information that helped her stand up to discriminatory treatment and protect her privacy. As she explained, she felt her position as a single mother with a precarious status to be “weak” and dependent, so having access to host society professionals helped her gain information that both gave her the agency to speak up and fortified her image as someone autonomous and competent, who “knows something” (Aisha).

Fatima emphasized the importance of having trustworthy information:


*Wrong information made [claimants think] there is no door open for me […] in this country [… and] they got depressed [that] way. […. So having access to] information gives me the strength that I’m not alone, there is a supporter.*


Fatima explained that having supportive host country connections and reliable information bolstered her self-efficacy in “solving problems” as she navigated resettlement.

A few participants eagerly shared that they distributed the information they learnt from the workshops and brought along other claimants to the workshop. Fatima reasoned that by bringing other claimants, she could help protect them from false information. In this way, having access to Welcome Haven became a tool for these participants to demonstrate their competence and support others.

Whereas information-based workshops empowered participants by equipping them with knowledge, art-based workshops empowered participants by encouraging creative self-expression. In the aftermath of social violence and migration, claimants frequently experienced self-doubt and a fractured sense of existential meaning. They noted that creative expression activities could help provide connection with their pre-migratory sense of self, make meaning of their journey, and give space for the expression of values that are fundamental to their identities. Fernanda described “the ability to say what I want to say in a way that I want it to be heard” in the creative expression workshops. Feeling “comfortable” to speak about her experiences at Welcome Haven helped her enhance the competence to self-express and develop a more “open and sociable” part of herself that was discouraged in her home country, which she believed would “facilitate [her] ability to integrate into the society here.”

Two participants mentioned that creative expression in the workshops reminded them to stay present-minded, which they believed to be essential for their wellbeing. Fernanda shared how she felt after making a collage that visualized her past, present, and future:


*[This activity] allowed me to shed the weight that I’ve been carrying since [my country of origin] [because] it forced me in a conscious way to let [my pre-migration experience] stay in the past, and to then be equipped to start my life here in Canada with a clear idea of where I want to go.*


Finally, mothers felt empowered when their family members were empowered at the workshop. Tenneh talked about “feeling proud” as she watched her young daughter explain her drawing to the group at the workshop: “she’s bold, […] she can explain herself, she can do something by her own.” Tenneh had previously expressed her hope for her daughter to be able to “stand on her own” and “fight for herself”, so seeing her daughter expressing herself was affirming for Tenneh’s parenting. Fernanda talked about witnessing her husband dance at the workshop, which she interpreted as a liberation from “the shame and repression he had in Mexico.” Mothers thus not only experience the value of the workshops for themselves but also for their families, particularly in perceiving their family members as self-expressive and empowered.

### 3.5. Enhanced Wellbeing

Most participants explicitly stated that attending Welcome Haven workshops had relieved their stress and ‘depression.’ One mother said that the workshops brought her “the breath of life.” A few participants highlighted the importance of laughter, which was often contrasted with their isolating state of uncertainty as they waited for their refugee determination hearing. These mothers described the possibility of workshops “chang[ing] tragedy [to] humour.” For Gloria, this transformation was inspired by dance workshops:


*You find a person who comes to dance in front of you with full of energy. You’ll ask yourself: Can I dance too? And that’s when you realize that yes, there are moments in life where you can tolerate. […] You don’t just have to cry. You can let go.*


Gloria continued to describe the sense of relief offered by the workshops:


*[The workshop] takes my mind off the current reality a little bit, I forget about the woman [crying] on the floor and I think about something else […] This is the only place where I smile.*


For Gloria, the workshops offered an important, but perhaps transitory, sense of wellbeing, distracting her from the seemingly intransigent “current reality” of precarious status, poverty, and domestic duties without social support. Having the workshop space became, for many, their only respite. A few mothers reported that they often felt more spirited returning home from the workshop. Tenneh shared that after a movement and dance workshop focused on dancing and movement, she recreated this activity outside of the workshop to lift her mood and to help her hypertension, which she worried about:


*You are tied in a house, no activities, […] you don’t do anything. So when I [was] at home the next day [after the workshop], I put some music, I try moving my body. […] I feel relief. I feel happy [and] I check my blood pressure, it was normal. So like, I see it is my stress. Whenever I’m working since that day, like if I want to clean up my apartment or trying to settle some things, I just put on the music.*


She had previously shared that she had escaped from gender-based violence in her family where music was forbidden. Thus, her adoption of music and dance in her own home was not just a step towards enhanced wellbeing but an agentic act of self-determination and identity making.

Overall, Welcome Haven offered a joyful atmosphere not only for the mothers we spoke to but also for their families. Several mothers also mentioned that participating in the workshops was “a very important family moment,” as both the parents and children were able to get out of their house and routine to engage in novel activities. A few mothers mentioned witnessing their children developing new creative hobbies, and for pre-school aged children without daycare access, the workshop provided a temporary space to develop socially and cognitively, as they were able to interact with other children. Fernanda shared that her children’s excitement in returning to the workshops showed her that “the workshops are a great place to be.” The intervention appeared to support wellbeing not just by improving mothers’ individual states of mind but also through a sense that their children were well-supported.

## 4. Discussion

This mixed-methods study sought to examine how self-efficacy, satisfaction of basic psychological needs, and adaptive stress influence asylum-seeking mothers’ subjective wellbeing during resettlement and to explore mothers’ experiences attending a community-based psychosocial program. To the best of our knowledge, this is the first study that attempts to understand the relationship between these psychological constructs in the context of a community-based psychosocial intervention, contributing to existing knowledge of “what works” for asylum-seeking families in these interventions. Importantly, need satisfaction and self-efficacy have never been compared head-to-head in this population, nor has adaptive stress been included in modeling with the aim to predict the relationship between agency and wellbeing. Our results illuminate the predictors of wellbeing in these mothers, suggesting that belonging and agency are important determinants. We also found that attending the intervention was associated with enhanced wellbeing and reduced adaptive stress, with the caveat that this relationship may not be causal. And finally, we found that adaptative stress has a critical role in mediating the effect of need satisfaction on wellbeing. Here, we explore the significance of these three main quantitative findings and link them with our qualitative results.

### 4.1. Need Satisfaction as Predictor of Wellbeing

As hypothesized, need satisfaction improved the prediction of adaptive stress and wellbeing above and beyond self-efficacy. Even though self-efficacy was initially found to be a significant predictor of wellbeing by itself, we theorized that need satisfaction would function as a better predictor of mothers’ subjective wellbeing and reduced resettlement stress because need satisfaction accounts for mothers’ sense of belonging through the factor of *relatedness* and autonomous values and desire through the factor of *autonomy*.

For the mothers in our study, *relatedness* appeared to be of central importance in shaping their wellbeing as emerged in our qualitative analysis. Participants expressed their appreciation of the communities they have fostered at Welcome Haven. This finding echoes previous research that found that providing asylum seekers opportunities to share with other newcomers their experiences coping with post-migration challenges [51] and to engage in cultural and social activities (such as sewing and cooking) [30,52] can promote social integration and strengthen resilience during resettlement [51]. Moreover, like others, we found that fostering positive individual relations between asylum seekers and host society members contributed to building mutual trust and respect [25,53,54]. The mental health benefits of fostering trusting relationships between citizens and displaced people may be particularly significant for refugee and asylum-seeking women, given the multifaceted forms of social exclusion and disadvantage they may face in a host country [55]. In this sense, we have shown that need satisfaction functions as a better predictor of mothers’ subjective wellbeing and reduced resettlement stress, for it accounts for mothers’ sense of belonging.

Claimant mothers shared that participating in activities that enhanced their capabilities and acknowledged their autonomy improved wellbeing. These findings point to the importance of implementing activities that not only make claimants feel capable (thus increasing their sense of self-efficacy) but also align with their values so that they will participate with self-determined motivation. Here, we theorize that need satisfaction, which takes into account women’s own desires and values in agentic action, is a key component that makes need satisfaction a better predictor of wellbeing than self-efficacy. In other words, it is not merely the ability to act that supports wellbeing but rather it is the ability to act in accordance with one’s beliefs and cultural values. This second finding suggests that involving asylum seekers themselves in intervention development to cultivate an approach that amplifies their expressed values, needs, and autonomy is critical. To this end, we recommend that intervention research with asylum seekers follows a participatory action approach and invites them as active collaborators to co-construct knowledge and interventions that are informed by their lived experience [56].

### 4.2. Frequent Program Attendance May Enhance Wellbeing

We also found that more frequent attendance at the Welcome Haven was associated with higher wellbeing and lower adaptive stress. These results suggest that Welcome Haven shows promise as an intervention to prevent the burden of adaptive stress and promote the wellbeing of asylum-seeking mothers. However, it is important to acknowledge that the relation may go in the reverse direction: better-adjusted asylum-seeking mothers with higher wellbeing and lower stress may make more visits, or a third variable may be influencing both the visits and adjustment (e.g., economic resources social integration).

### 4.3. Mediating Effect of Adaptive Stress

Our explorative mediational analysis demonstrates the key role of adaptive stress in attenuating the relationship between need satisfaction and wellbeing: once adaptive stress is accounted for, need satisfaction no longer significantly predicts wellbeing. In other words, better satisfaction of mothers’ psychological needs has the effect of reducing adaptive stress, which in turn leads to better wellbeing. This also means that for asylum-seeking mothers, the satisfaction of their basic psychological needs itself is not enough. Instead, the reduction of adaptive stress plays a greater role in supporting wellbeing. Therefore, implementing programs such as Welcome Haven that provisionally satisfy basic psychological needs may be insufficient to sustainably enhance wellbeing in the face of ongoing systemic inequality and marginalization. Similarly, Chase and Rousseau [25] cautioned that a drop-in day centre for asylum seekers could not “restore a complete sense of community among [asylum seekers] whose future remains ‘suspended’” (p. 55). Indeed, resettlement stressors are interconnected and pervasive for asylum seekers during post-migration and appear to predict worse mental health [1,2]. The reparation of post-migration stressors demands that host societies take “a broader psychosocial perspective that recognizes a coordinated, multi-sectorial approach” [22] (p. 246) as well as the development of policy that protects asylum claimants’ mental health through minimization of uncertainty, social and economic exclusion, and hostile conditions of reception. Our study contributes to the growing literature that advocates for addressing adaptive stress as a key determinant of asylum seekers’ mental health.

### 4.4. Qualitative Findings

The qualitative results may help illuminate the mechanisms by which the Welcome Haven satisfied mothers’ basic psychological needs and enhanced wellbeing. Mothers expressed satisfaction through social connections with other asylum seekers and host society members, as well as a sense of competence and autonomy through access to information and engagement in creative self-expression. Interestingly, mothers emphasized that they only experienced a sense of enhanced wellbeing when *all* their family members benefitted from the workshop, suggesting that women endorse a collectivist conceptualization of wellbeing rather than one focusing on the individual self. Mitschke et al. [57] point out that most refugees prefer group interventions to individual counseling, the predominant Western intervention model. IASC [34] similarly recommends a staged, pyramidal approach to intervention for people affected by crisis. They advise that psychosocial programs that support family and community recovery be prioritized and that specialized, clinical interventions are the focus for only a small portion of the population. The Welcome Haven provides preventative mental health support to larger groups while allowing the identification of more vulnerable parents and children who require more specialized mental health services. Our study provides more evidence of the effectiveness of these community-based psychosocial programs in protecting asylum-seeking mothers’ wellbeing and supports this pyramidal approach. 

Finally, our qualitative results revealed that participants derived a sense of competence and autonomy through access to reliable information and services. Their liaison with service providers at the program not only helped claimants attain important resources and advocate for their needs but also supported them in their roles as mothers and in establishing a sense of mastery in navigating resettlement. First, this result highlights the need for the preservation of these venues for psychosocial support and integration services, which are usually facilitated by community-based groups [58], whose work was neglected during the pandemic [59]. Second, we point to the reparation of the ADAPT pillars of role/identity and sense of existential meaning [24] via protecting and promoting pathways to education and employment for the reparation of the pillars of role and identity and sense of existential meaning. Specifically for Quebec, Cleveland et al. [58] listed a number of ways the government could help claimants transition and reprise meaningful roles in host societies, such as systematically assessing asylum seekers’ skills, qualifications, and employment objectives in order to provide them with further training or match them with existing labour needs. For asylum-seeking mothers, access to subsidized daycare is particularly crucial to their ability to take language classes, and access education or training. Overall, it is essential to set up welcoming services that will facilitate asylum seekers’ resettlement and integration to prevent social exclusion and inequity.

### 4.5. Limitations

The limitations of the current study primarily lie in the small sample size of the quantitative portion of the study. Second, we could only draw correlational results because of the lack of pre–post comparisons in program participation as a result of the drop-in nature of the program. For both limitations, the addition of qualitative methods helped us enrich the quantitative data and provide a more comprehensive view [60]. Nevertheless, better-controlled experiments in the future will allow more conclusive evidence of intervention efficacy. Sampling challenges for quantitative studies with refugees and asylum seekers have been well-documented [61]. Given that our sample was recruited from an intervention program, which introduces the possibility of selection bias, the generalization of our quantitative data should be carried out with caution. Finally, the current study used shortened versions of most scales. In the case of self-efficacy, this resulted in lower internal consistency and might have biased the results. Shortened scales were, however, necessary considering our intention to avoid over-burdening claimant participants, especially during program activities that were designed to relieve stress.

### 4.6. Conclusions

The current study highlights asylum-seeking mothers’ need for relatedness and belonging during resettlement and calls for implementation and evaluation of community-oriented support that is not solely based on an individualistic model of mental health. As we found adaptive stress to be a key determinant of claimant mothers’ wellbeing, we caution that implementation of psychosocial interventions risks being superficial if the pervasive resettlement stressors are not addressed through systemic policy changes. With increased global displacement, a better understanding of these psychosocial frameworks and ecological interventions is imperative to assisting asylum seekers’ post-migratory adjustment and ensuring health equity.

## Figures and Tables

**Figure 1 ijerph-20-07076-f001:**
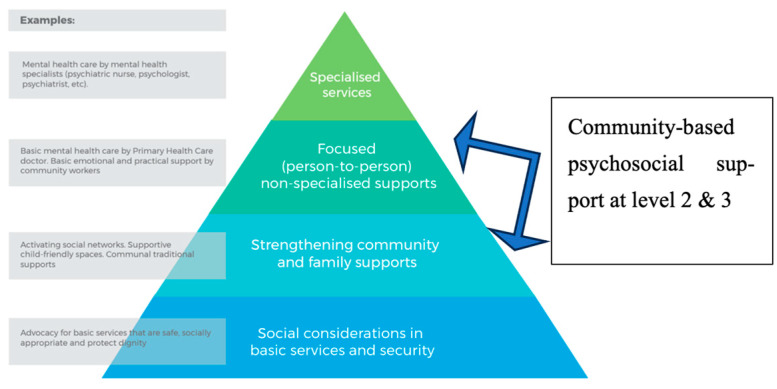
Intervention pyramid for mental health and psychosocial support in emergencies.From: IASC guidelines on mental health and psychosocial support in emergency settings. Interagency Standing Committee. 2007. Available online: http://www.who.int/mental_health/emergencies/guidelines_iasc_mental_health_psychosocial_june_2007.pdf (accessed on 18 October 2023).

**Figure 2 ijerph-20-07076-f002:**
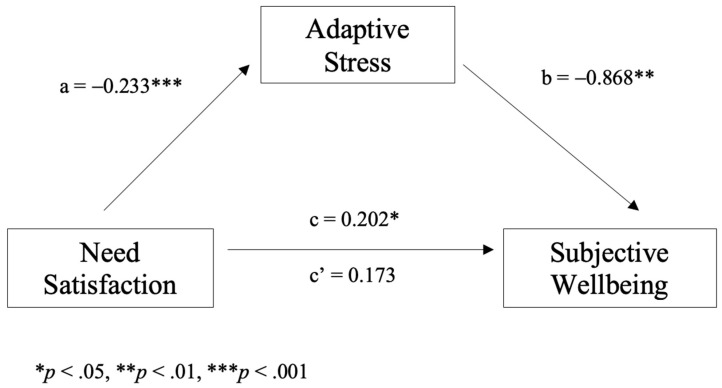
The simple mediation effect of adaptive stress in the relation between need satisfaction and subjective wellbeing.

**Table 1 ijerph-20-07076-t001:** Summary of measurement scales.

Name	Items	Likert Scale	Cronbach’s Alpha
Satisfaction With Life Scale (shortened)	In most ways my life is close to ideal. The conditions of my life are excellent. I am satisfied with my life. If I can live my life over, I will change almost nothing.	7-point Likert scale: (1) “not at all true” to (7) “very true”	0.75
Mood Report	Joyful Unhappy Worried/Anxious Enjoyment/Fun Depressed Pleased Happy Angry/Hostile Frustrated	7-point Likert scale: (1) “not at all” to (7) “extremely”	0.88
Basic Psychological Need Satisfaction Scale (shortened)	I feel I do things well. I enjoy spending time with the people around me. I feel I have a choice about important aspects of my life. I feel like people around me think I was good at things. I feel free to express myself. I feel like the people around me like me and care about me.	4-point Likert scale: (1) “not at all” to (4) “extremely”	0.87
General Self-Efficacy Scale (shortened)	It is easy for me to accomplish my goals. I am confident I could deal with unexpected events. I can stay calm when facing difficult situations.	4-point Likert scale: (1) “not at all true” to (4) “exactly true”	0.63
Adaptive Stress Index: Role and Identity; Existential Meaning.	I am frustrated because of the communication difficulties I have been having. I feel lost as if I have lost control of my life. I am frustrated because I am not able to contribute to the community. I have lost a sense of purpose or meaning in my life. I have difficulty trying to overcome cultural barriers. I have trouble making sense of the bad thing that happened in my life. I feel frustrated because I have to rely on others I have lost the sense of autonomy and control in my life I feel frustrated because I do not have access to religious or spiritual practices	4-point Likert scale: (1) “not at all” to (4) “extremely”	0.81

**Table 2 ijerph-20-07076-t002:** Sociodemographic characteristics of the sample (N = 40).

Sociodemographic Variable	*n*	% or *M* (*SD*)
**Age**	40	35.4 (7.2)
**Highest level of education**		
Primary school	0	0
Secondary school	14	35.0%
University and above	25	62.5%
**Civil status** 1—Single 2—Married	2 34	5.0% 59.6%
3—Separated	1	2.5%
4—Divorced	1	2.5%
5—Widowed	0	0%
6—Prefer not to answer	1	2.5%
**No. of children** **Length of stay (months)** **Program attendance**	40 38 39	2.4 (1.0) 4.6 (8.6) 4

**Table 3 ijerph-20-07076-t003:** Hierarchical regression analysis on predictors for subjective wellbeing.

Predictor	B	SE B	ß	t	*p*	VIF	ΔR^2^	Adj R^2^
**Model 1**								0.087
Constant	4.031	0.302		13.360	<0.001			
Length of stay	−0.068	0.030	−0.507	−2.250	0.031	2.060		
Program attendance	0.313	0.150	−0.471	2.091	0.044	2.060		
**Model 2**							0.095	0.182
Constant	2.050	0.924		2.218	0.033			
Length of stay	−0.054	0.029	−0.402	−1.838	0.075	2.159		
Program attendance	0.284	0.142	0.427	1.994	0.054	2.077		
Self-efficacy	0.629	0.279	0.344	2.253	0.031	1.057		
**Model 3**							0.128	0.310
Constant	0.888	0.952		0.934	0.357			
Length of stay	−0.063	0.027	−0.473	−2.337	0.026	2.197		
Program attendance	0.340	0.132	0.512	2.570	0.015	2.130		
Self-efficacy	0.317	0.218	0.173	1.125	0.269	1.273		
Need satisfaction	0.378	0.140	0.408	2.700	0.011	1.223		

*N* = 38.

**Table 4 ijerph-20-07076-t004:** Hierarchical regression analysis on predictors for adaptive stress.

Predictor	B	SE B	ß	t	*p*	VIF	ΔR^2^	Adj R^2^
**Model 1**								0.048
Constant	2.122	0.151		14.014	<0.001			
Length of stay	0.017	0.015	0.271	1.154	0.256	2.079		
Program attendance	−0.143	0.074	−0.451	−1.922	0.063	2.079		
**Model 2**							−0.01	0.038
Constant	2.499	0.490		5.103	<0.001			
Length of stay	0.015	0.015	0.231	0.959	0.345	2.170		
Program attendance	−0.138	0.075	−0.437	−1.848	0.074	2.091		
Self-efficacy	−0.119	0.147	−0.136	−0.810	0.424	1.056		
**Model 3**							0.26	0.298
Constant	3.268	0.469		6.974	<0.001			
Length of stay	0.021	0.013	0.331	1.596	0.120	2.209		
Program attendance	−0.176	0.065	−0.557	−2.722	0.010	2.146		
Self-efficacy	0.084	0.137	0.096	0.611	0.546	1.264		
Need satisfaction	−0.247	0.068	−0.561	−3.639	<0.001	1.220		

*N* = 37.

**Table 5 ijerph-20-07076-t005:** Demographic information of interview participants.

Participant Code	Region of Origin	Length of Resettlement (Months)	Marital Status	Number of Children	Education Level	Language
Gloria	Central Africa	7	Married	2	Professional	French
Cassandra	Caribbean	16	Married *	1	Professional	French
Aisha	East Africa	36	Single	2	Highschool	English
Fatima	East Africa	4	Single	1	Professional	English
Mary	West Africa	27	Single	2	Professional	English
Tenneh	West Africa	2	Single	2	Highschool	English
Rosine	Central Africa	5	Married *	2	Professional	French
Adriana	South America	5	Married	3	Professional	Spanish
Selma	West Africa	3	Married	2	Professional	English
Daniela	South America	1.5	Married	2	Professional	Spanish
Divine	Central Africa	2	Married	3	Highschool	French
Fernanda	Central America	4	Married	3	Professional	Spanish
Rosa	South America	3	Married	2	Professional	Spanish

* Indicates married but with partner still in country of origin.

## Data Availability

Questionnaire results and excerpts of de-identified transcripts relevant to the study may be made available upon request, excluding any that might compromise participant confidentiality.

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
