# Peer review of "The Relationship between Wellbeing, Self-Determination, and Resettlement Stress for Asylum-Seeking Mothers Attending an Ecosocial Community-Based Intervention: A Mixed-Methods Study"

_ijerph, 2023, doi:10.3390/ijerph20227076_

Round 1

Reviewer 1 Report

Comments and Suggestions for Authors

Dear authors,

I would like to express my appreciation for your diligent work on this manuscript. It is evident that the research topic holds significant importance, and I commend your efforts in integrating relevant prior research into your study.

I do have some constructive feedback and recommendations to help enhance the clarity and impact of your work:

Major Recommendations:

1. The term 'Community-based psychosocial support programs' is used in the manuscript without a clear explanation of its specific characteristics and how it differs from alternative support programs. To enhance the clarity of your research, consider providing a detailed explanation of this term and comparing it to other types of support programs to highlight its unique features and value.

2. The study's aims and research objectives could benefit from greater clarity. Specifically, the rationale for selecting the relationships between the mentioned constructs and the hypotheses you've outlined could be better justified. It's essential to provide a clear and compelling rationale for your research focus and objectives to establish their significance.

3. Throughout the literature review, there is an absence of specific mention of mothers as the focus of your study. Given the empirical focus on mothers, it would be beneficial to include a paragraph in the literature review that discusses the rationale for this focus in contrast to other demographic groups or family members. Additionally, consider addressing the specific context of mothers with children, which appears to be a significant factor in your research.

3. The use of self-efficacy as a measure in your study is valuable; however, it appears that in some instances, the construct is downplayed in the literature review (e.g. "Self efficacy, which does not include a measure… is thus a potentially less potent predictor…" or later "…post-migratory stressors were found to be much more significant… compared to self efficacy…".). Since you have included this measure in your work, consider providing a balanced perspective on its relevance and importance within the context of your study.

4. The mediation model presented in Figure 1 should be theoretically well-supported to justify its inclusion in the study. As mentioned earlier, the lack of theoretical justification for the hypotheses weakens the overall model. Additionally, it appears that the mediation effect in the model is relatively small (c = 0.200, c' = 0.215), which questions its practical significance.

5. In the qualitative section, please ensure that quotation marks are consistently used where necessary. These marks are essential for clarity and proper citation of participants' statements.

Minor Recommendations:

1. There is a minor typo in the manuscript: "When applied to trauma-affected individuals," where the article 'a' should be removed for grammatical correctness.

2. The manuscript would benefit from a more explicit explanation of how the quantitative and qualitative components of the study complement each other within the mixed-methods approach.

3. Please verify the inclusion of a correlation table for the main study variables, as it appears to be missing.

4. Consider measuring PANAS scale items (positive and negative affect) separately rather than as a composite index that includes life satisfaction. This approach may provide a more nuanced understanding of the data and enhance the interpretability of the results. It is less acceptable in research to combine satisfaction with affective experience.

5. The standardization (z-scoring) of control variables should be justified in the manuscript. Standardization may not be necessary unless there are specific reasons for this transformation.

6. Given the small number of participants, consider using a stepwise model for covariates to ensure a more robust and interpretable analysis. Variables that were found to be non-significant (e.g., age, education, number of children) may be removed from the models to simplify the analysis. Additionally, the variable "civil status" may need to be reconsidered based on its distribution.

7. Clarify whether the variable "education" refers only to completed education or includes individuals who are currently pursuing their studies.

I hope you find these recommendations valuable for enhancing the quality and impact of your research. Your commitment to addressing these points will contribute significantly to the overall strength of your manuscript.

Reviewer 2 Report

Comments and Suggestions for Authors

This mixed-methods study sought to quantitatively examine how self-efficacy, satis-474 faction of basic psychological needs, and adaptive stress influence asylum-seeking moth-475 ers’ subjective wellbeing during resettlement, and qualitatively explore mothers’ experi-476 ences attending Welcome Haven. However, there is still room for further improvement in this paper, and the following suggestions are for reference only.

1. The logic of literature review in this paper is weak, so it is suggested that the author reorganize the research topic and quote as many authoritative literatures in this field as possible.

2. There are too many empirical regression results in the text, which can be clearly obtained from tables or charts. It is suggested that the author only show the necessary data and regression results to enhance the readability of the paper.

3. It is suggested that before the empirical analysis, the author should have a clear theoretical analysis to show the logical relationship among self-efficacy, self-determination, happiness and resettlement pressure.

4. Many cases found in the course of investigation are cited in this paper. It is suggested that the author should clearly introduce its background and source when citing, so as to enhance the rigor of the paper.

Round 2

Reviewer 1 Report

Comments and Suggestions for Authors

Dear Authors,

I would like to extend my appreciation for your diligent efforts in addressing the required corrections in the initial version of the manuscript. Your responsiveness is greatly valued.

I have reviewed the revised version and have a few additional comments that I believe will help enhance the overall quality of the paper.

Major comments:

The value of this research, as currently stated, is unclear. The statement, "While previous research has looked at psychological theories related to social support, acculturation, and empowerment (Shaw & Funk, 2019), few studies have examined any other psychological models that could facilitate the flourishing of asylum-seeking mothers in resettlement," lacks clarity. It's not clear what is meant by "any other psychological models," and the intended contribution of the paper is vague.

Additionally, later on you mention agency and belonging as two important concepts in your investigation, while earlier in text you mentioned that "previous research has [already] looked at psychological theories related to social support, acculturation, and empowerment" – so again, the unique contribution of this paper is not clear, since studying social support and empowerment is not very different from researching the topics of belonging and agency, as far as I could consider these matters. The goal, importance, and research question of the study are not well-defined.

Minor comments

In the methods section, the explanation for using both quantitative and qualitative data is somewhat general and lacks depth. It would be beneficial to provide a more specific explanation of how each research method contributes to the study's goals and how they complement one another in achieving a comprehensive understanding of the research topic.
